# Unraveling the synergistic effects of Cu-Ag tandem catalysts during electrochemical CO$_2$ reduction using nanofocused X-ray probes

Marvin L. Frisch [1,6], Longfei Wu [1,2,6], Clément Atlan[3,4], Zhe Ren[5], Madeleine Han[3], Rémi Tucoulou[3], Liang Liang[1], Jiasheng Lu [1], An Guo[1], Hong Nhan Nong [1], Aleks Arinchtein[1], Michael Sprung [5], Julie Villanova [3], Marie-Ingrid Richard [3,4] & Peter Strasser [1] ✉

Controlling the selectivity of the electrocatalytic reduction of carbon dioxide into value-added chemicals continues to be a major challenge. Bulk and surface lattice strain in nanostructured electrocatalysts affect catalytic activity and selectivity. Here, we unravel the complex dynamics of synergistic lattice strain and stability effects of Cu-Ag tandem catalysts through a previously unexplored combination of in situ nanofocused X-ray absorption spectroscopy and Bragg coherent diffraction imaging. Three-dimensional strain maps reveal the lattice dynamics inside individual nanoparticles as a function of applied potential and product yields. Dynamic relations between strain, redox state, catalytic activity and selectivity are derived. Moderate Ag contents effectively reduce the competing evolution of H$_2$ and, concomitantly, lead to an enhanced corrosion stability. Findings from this study evidence the power of advanced nanofocused spectroscopy techniques to provide new insights into the chemistry and structure of nanostructured catalysts.

The capture and subsequent electrocatalytic conversion of CO$_2$ gas (eCO$_2$RR) into value-added products, such as fuels, syngas, or alcohols, using renewable electricity represents one of the most attractive routes to establish a sustainable, circular economy, and mitigate anthropogenic CO$_2$ emissions[1–3]. For a cost-competitive eCO$_2$RR process on an industrial scale, durable and selective electrocatalysts are indispensable. Product selectivity during eCO$_2$RR is mainly affected by the properties of the electrocatalytic materials employed at the cathode, where the reduction proceeds, and their microenvironment[4]. Cu is the only monometallic candidate being capable of producing not only C$_1$ compounds, such as carbon monoxide (CO) or methane (CH$_4$), but also C$_{2+}$ products, such as ethylene (C$_2$H$_4$) or ethanol (CH$_3$CH$_2$OH)[1,5].

Despite the complexity of the observed mechanistic eCO$_2$RR pathways, scaling relations were identified, which can serve as the basis for rational catalyst design[1,6,7]. Based on the differences in the CO binding energy to a metal surface, the volcano plot for eCO$_2$RR was used to explain the enhanced selectivity for Cu-based materials toward C$_{2+}$ products. Contrary to Cu, Ag is found on the weak binding side of the apex of the volcano, favoring CO gas desorption[7,8]. Recently, Berlinguette and co-workers outlined the potential of strain engineering in electrocatalysis to enhance eCO$_2$RR activity and selectivity by breaking linear scaling relations[7]. Yet, the main challenge is to disentangle ligand or strain effects from geometric effects, as clearly pointed out in a comprehensive review article by Nitopi et al.[1] To date, there exist only a few

[1]Department of Chemistry, Chemical Engineering Division, Technische Universitaet Berlin, Str. des 17. Juni 124, 10623 Berlin, Germany. [2]Alexander von Humboldt Foundation, Jean-Paul-Str. 12, 53173 Bonn, Germany. [3]ESRF, The European Synchrotron, 71 Avenue des Martyrs, Grenoble 38000, France. [4]CEA Grenoble, IRIG/MEM/NRX, Université Grenoble Alpes, Grenoble 38054, France. [5]Deutsches Elektronen-Synchrotron (DESY), Notkestr. 85, 22607 Hamburg, Germany. [6]These authors contributed equally: Marvin L. Frisch, Longfei Wu. ✉e-mail: pstrasser@tu-berlin.de

theoretical and experimental studies providing clear evidence for strain-induced changes in eCO$_2$RR activity or selectivity. Different approaches were presented in the literature to deliberately induce strain in Cu-based eCO$_2$RR catalysts, e.g. via epitaxial thin-film growth[9,10] or bimetallic nanoparticle (NP) formation[11,12]. Lei et al.[13] point out the relevance of in situ/operando characterization methods in a recent contribution. (Hydr)oxide-derived Cu catalysts were investigated by a combination of operando X-ray diffraction (XRD) and operando Raman spectroscopy for the identification of active sites which would remain hidden via ex situ analysis. Importantly, the study discloses distinct variations in the degree of lattice strain depending on the choice of the Cu precursor. According to their findings, the tensile strain would contribute to the formation of C$_1$ products. Contrarily, Kim et al.[10] concluded a suppression of the latter for tensile-strained Cu(001) surfaces. Despite controversial hypotheses, strain engineering has emerged as an effective tool to tune eCO$_2$RR selectivity.

In this contribution, we synthesize and characterize well-defined bimetallic Cu-Ag tandem NP model catalysts with controlled varying molar Ag fraction, before we investigate their structure-activity-selectivity relations in situ under eCO$_2$RR conditions. To achieve this, we develop, validate, and utilize a combined X-ray nanoprobe methodology (Fig. 1) that is capable of tracking the evolution of the chemical (oxidation) states, as well as the three-dimensional atomic displacement distribution and local strain in individual Cu-Ag NPs via in situ nanofocused XAS (nano-XAS) and BCDI (nano-BCDI), respectively. Building on our previous work[5], we use the obtained structure-activity-selectivity relations to formulate design guidelines of tandem nano-catalysts with enhanced total (C$_1$ and C$_{2+}$) CO$_2$ reduction product yields, while suppressing the competing hydrogen evolution reaction (HER) as a result of the presence of phase-segregated Ag domains. While this study is not focused on boosting any single CO$_2$ reduction product yield, it considers total CO$_2$ reduction product yield and also provides evidence that Cu-Ag NP catalyst systems offer a, previously overlooked, significantly improved corrosion resistance compared to monometallic Cu NPs. Together, the presented methodological approach to characterizing Cu-Ag tandem NPs advances our fundamental understanding of the complex eCO$_2$RR by providing evidence for identifying catalytically active sites. More broadly, this contribution highlights the power of the combination of nano-XAS/nano-BCDI for the characterization of individual nanoscale material entities.

## Results

To utilize the proposed combined in situ nanofocused X-ray absorption and diffraction methodology, a synthesis route affording well-dispersed, bimetallic Ag-Cu NPs with an average size of at least 100 nm, and similar composition and crystallinity was developed. The isolated model catalyst NPs were prepared via sputter-coating and subsequent dewetting on polished, electrically conductive glassy carbon (GC) substrates (Fig. 2a and Supplementary Fig. 1). Annular dark-field scanning transmission electron microscopy (ADF-STEM) coupled with energy-dispersive X-ray spectroscopy (EDX) indicates the formation of a large, Cu-rich domain alongside a small, Ag-rich domain for Ag-modified Cu NPs (Fig. 2b–d and Supplementary Fig. 2). By the combination of STEM-EDX and selected area electron diffraction (SAED) analyses, a gradient in composition at the interface between Cu and Ag is found (Supplementary Figs. 2 and 3). The phase-segregated morphology is further corroborated by SEM-EDX elemental mappings (Fig. 2g, h). In addition, improved morphological stability for Ag-modified Cu catalyst NPs compared to monometallic Cu NPs (Fig. 2e, f and Supplementary Figs. 4–6) can be claimed. XRD analysis (Fig. 2i and Supplementary Fig. 5) provides further evidence for the formation of phase-segregated domains of pure Cu and Ag in the tandem catalyst, which can be attributed to the positive enthalpy of Cu-Ag solid solutions[5,14].

To assess the stability of the synthesized NPs, eCO$_2$RR tests were conducted in a three-electrode H-cell setup. The Cu-Ag tandem catalysts were tested with a focus on selectivity and durability. Results revealed that monometallic Cu NPs undergo severe restructuring and partial dissolution/corrosion after 6 h eCO$_2$RR, whereas bimetallic Cu-Ag catalysts show improved durability without any pronounced sintering or particle detachment from the electrode surface (Supplementary Figs. 4, 6). Beside a strong adhesion of the NPs to the substrate, the absence of pronounced coalescence or sintering processes is critical for meaningful investigations using nanofocused X-ray probe techniques (cf. Fig. 2j). Reaction-induced morphological changes were previously reported to alter the eCO$_2$RR product selectivity, e.g., by variations in the surface atomic coordination number or the exposed facets[5,15,16].

Note that comparative EDX analyses of the pristine and tested bimetallic electrocatalysts indicated not only a very similar particle morphology but also an almost identical bulk composition (Fig. 2g, h and Supplementary Fig. 6). Both Cu- and Ag-rich domains remain discernible after the chronoamperometric durability test. A comparable stabilization effect at a moderate cathodic bias (−0.8 V$_{RHE}$) was

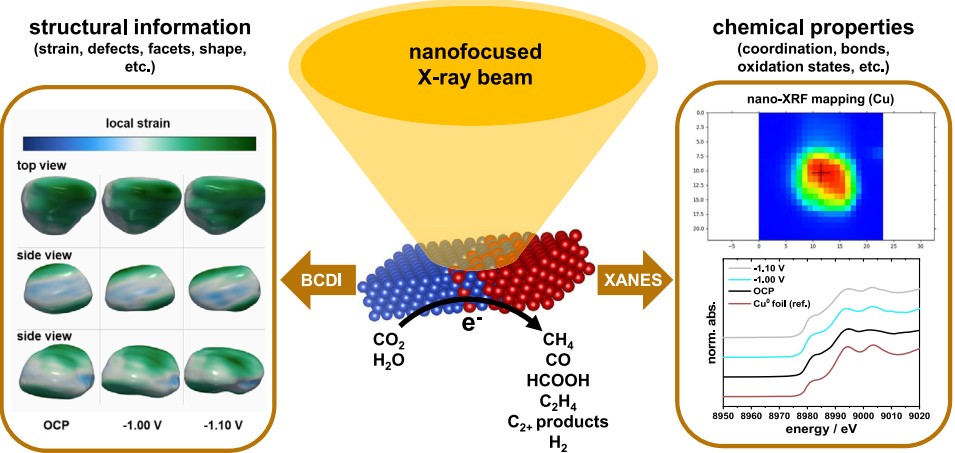

**Fig. 1 | In situ nanofocused X-ray characterization as a powerful yet underutilized tool for the structural and chemical analysis of nanoscale objects.** Here, a combination of nanoprobe techniques is applied for the identification of active sites in bimetallic Cu-Ag catalysts and the exploration of structure-reactivity relations during eCO$_2$RR.

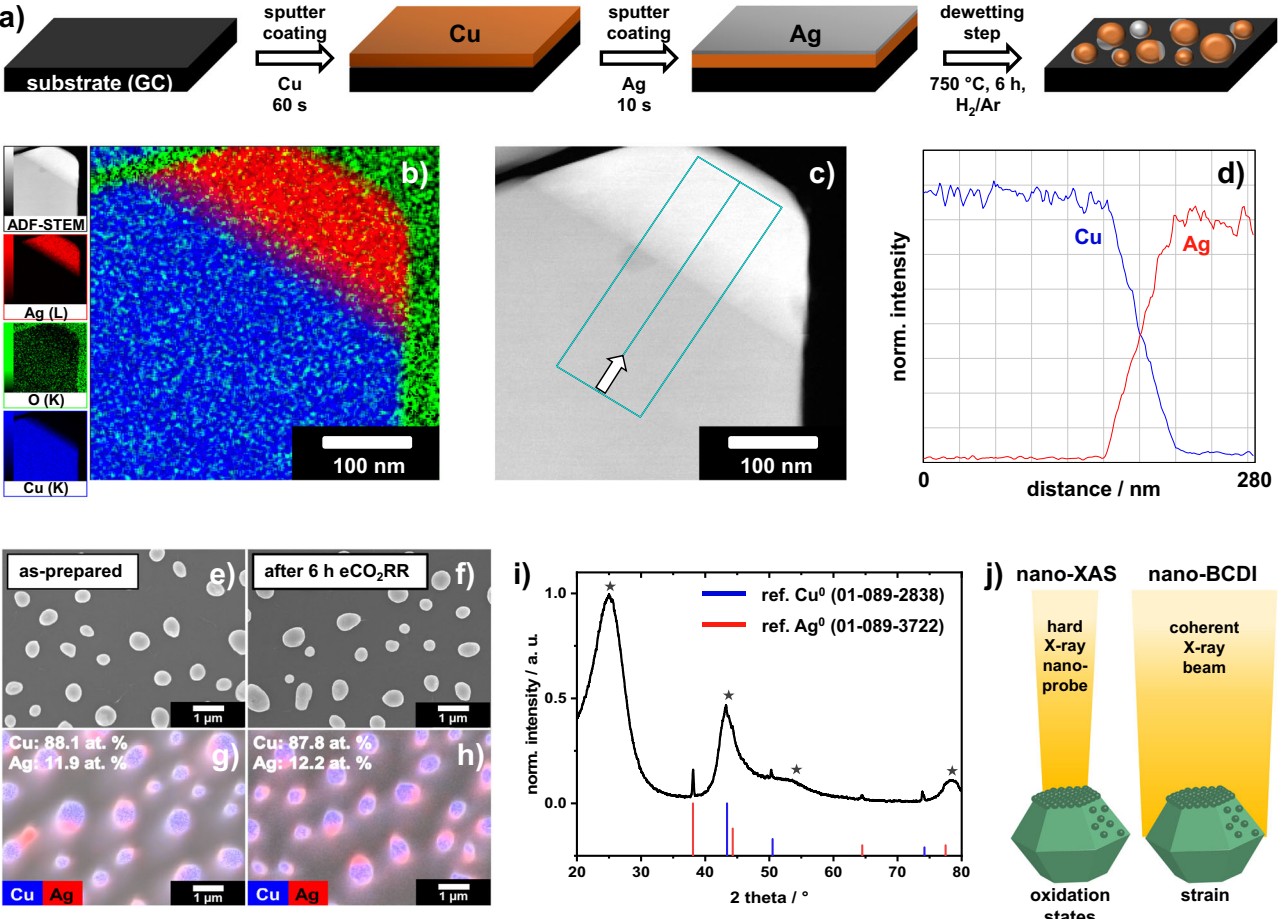

**Fig. 2 | Characterization of the Cu-Ag model catalyst. a** Schematic illustration of the developed synthesis route for isolated Cu-Ag NPs on an electrically conductive GC substrate. **b** Representative ADF-STEM-EDX mapping of a single, bimetallic NP (Cu$_{0.88}$Ag$_{0.12}$ catalyst). **c** Corresponding ADF-STEM image revealing the Cu-Ag interface and **d** the elemental distribution obtained from the EDX line-scan (**c**). **e, f** Representative SEM images of Cu$_{0.88}$Ag$_{0.12}$ and **g, h** corresponding SEM-EDX elemental mappings before and after eCO$_2$RR in an H-cell setup in CO$_2$-saturated 0.1 M KHCO$_3$ electrolyte at room temperature. **i** GI-XRD analysis of the spent Cu$_{0.88}$Ag$_{0.12}$ catalyst revealing the presence of segregated Cu$^0$ and Ag$^0$ phases, according to the corresponding reference patterns shown as blue and red vertical bars, respectively. Broad reflections indicated by dark gray asterisks can be attributed to the GC substrate. **j** Nanofocused X-ray characterization techniques for the identification of active sites via combined structural and chemical analyses under reaction conditions.

---

observed in our previous contribution[5] for Cu-Ag NPs synthesized via ligand-assisted electrophoresis. Importantly, findings from the present study indicate a stabilization effect of the NPs against coalescence and corrosion even for low Ag contents (see Supplementary Fig. 6). Obviously, undesirable dissolution-redeposition processes[17]—leading to particle growth at sufficiently cathodic bias—are restrained by the presence of predominantly isolated Ag domains in the NPs (cf. Fig. 2b).

Besides improved durability, Cu-Ag tandem catalysts reveal an enhanced eCO$_2$RR selectivity toward C$_1$ products, particularly CO and CH$_4$ (Fig. 3a and Supplementary Fig. 7). The introduction of Ag species suppresses the kinetically favored HER side-reaction to a large extent, as previously reported in the literature[18]. The bimetallic nature is expected to affect both catalytic activity and selectivity, as described for mono-[10,19] and bimetallic[4,7] Cu-based eCO$_2$RR catalysts before. As a first step, we establish in situ nanofocused XANES to track the chemical states of individual NPs. Advantageously, with a beam size of ~80 × 100 nm (H × V), hard X-ray nanoprobe techniques at the ID16B beamline of the European Synchrotron Radiation Facility (ESRF, Grenoble, France) provide a superior spatial resolution for time-resolved nano-XAS measurements under operating conditions. Figure 3b schematically illustrates a customized cell built for potential-dependent eCO$_2$RR measurements in liquid electrolyte (see Supplementary Fig. 8 for further details). A previously established nano-XRF technique[20,21] was used for the mapping of the NPs. As depicted in Supplementary Fig. 9, Cu-specific elemental distribution maps were obtained from the acquired nano-XRF data with submicron resolution. Based on these elemental maps, the location-specific chemical structure can be resolved for an individual particle. At open circuit potential (OCP), the results imply a higher average oxidation state, i.e. the presence of Cu$^{I/II}$ species in close vicinity of the interface to the liquid electrolyte (see Supplementary Fig. 9)[17,22]. Evaluations of the acquired in situ XANES series unambiguously reveal the predominant existence of metallic Cu$^0$ species under cathodic bias, as indicated by the Pourbaix diagram of Cu[1,13]. Here, it has to be mentioned that no conclusions about the exact composition or oxidation states at the surface of a particle can be drawn, which lies beyond the scope of this work and would require a more surface-sensitive technique. Nonetheless, XANES spectra collected at the particle rim show similar features to those from the center (Supplementary Figs. 10–13), not providing any clear evidence for oxidized bulk Cu$^{x+}$ species. Note that the stable presence of Cu$^I$ species during eCO$_2$RR remains a subject of debate and advanced concepts were recently proposed to control the ratio of Cu$^0$/Cu$^I$ species for an enhanced C$_{2+}$ product selectivity[23–26].

As a next step, in situ nano-BCDI experiments were carried out to track the atomic displacement distribution and strain (here, along the [002] direction) as a function of the applied potential in an individual NP (Figs. 4 and 5 and Supplementary Fig. 14). For the evaluation of the obtained diffraction patterns (Fig. 4) and a reliable reconstruction of

**Fig. 3 | Potential-dependence of eCO₂RR selectivity, in situ nano-XAS setup, and in situ XANES spectra of the Cu-Ag model catalyst. a** Potential-dependent Faradaic efficiencies of $Cu_{0.88}Ag_{0.12}$ in eCO₂RR in an H-cell setup in CO₂-saturated 0.1 M KHCO₃ electrolyte. **b** Schematic illustration of the developed nano-XAS setup at ID16B-ESRF beamline for the location- and potential-resolved acquisition of XANES data. **c** In situ Cu-K-edge XANES spectra (normalized) of individual

$Cu_{0.88}Ag_{0.12}$ NPs during eCO₂RR and **d** corresponding first derivatives (normalized). As a reference, a $Cu^0$ metal foil was investigated in a dry state without any applied potential. Note that all potentials were converted to the RHE scale and 100% iR-compensated. Error bars in **a** represent standard deviations for triplicate experiments.

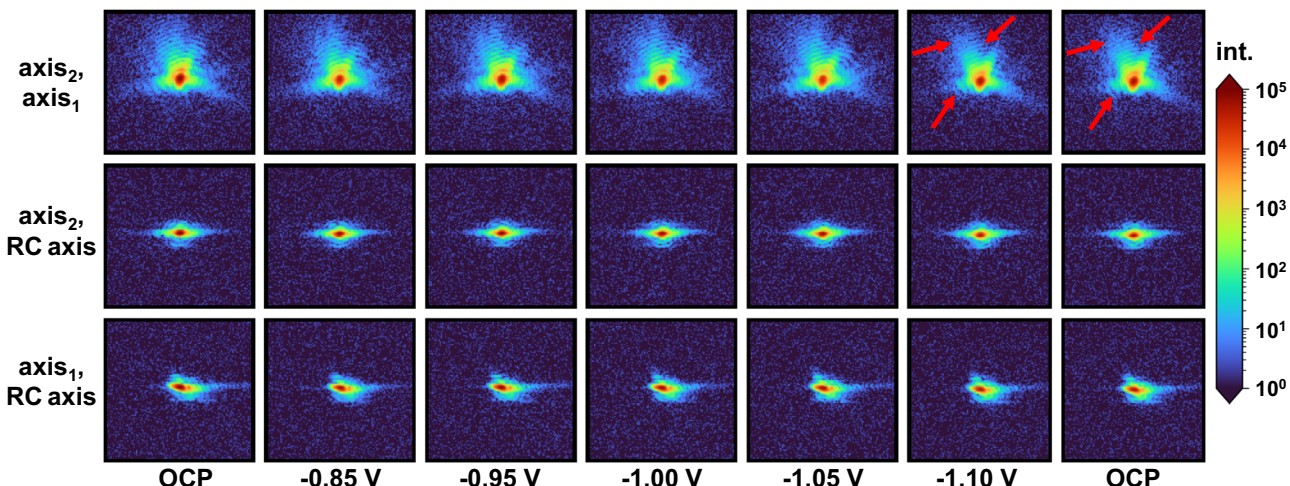

**Fig. 4 | In situ nano-BCDI for strain analysis on individual NPs during eCO₂RR using coherent X-rays at the P10 beamline (PETRA III) at Deutsches Elektronen-Synchrotron (DESY, Germany).** Representative diffraction patterns acquired at varying potentials on an individual $Cu_{0.88}Ag_{0.12}$ NP are illustrated. Red arrows highlight slight changes in the acquired fringe patterns observed under more

negative potentials. Rocking curve (RC) parameters are given in the SI. Axis₁ and axis₂ correspond to the horizontal and vertical axes of the bi-dimensional detector, respectively. Note that all potentials were converted to the RHE scale and 100% iR-corrected.

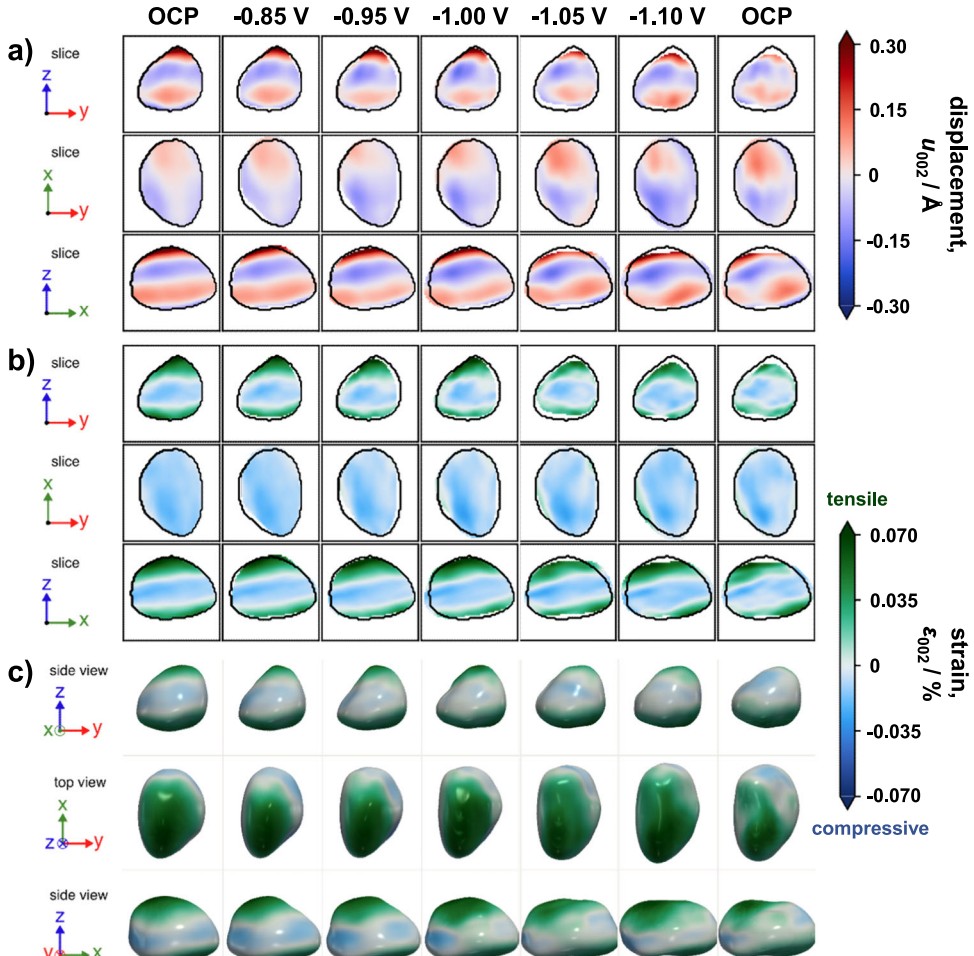

**Fig. 5 | In situ nano-BCDI reconstructions of an individual eCO$_2$RR catalyst NP.**
**a**, **b** 2D, and **c** 3D reconstructions (real space) were obtained by applying a phase retrieval algorithm on the measured coherent X-ray diffraction pattern data. The size of the NP is ~450 nm. **a**, **b** the initial shape of the reconstructed crystal (at OCP) is shown as a guide to the eye (see black lines). Z is along the particle's height, i.e., perpendicular to the substrate. The atomic displacement ($u_{002}$) and the strain ($\varepsilon_{002}$) along the [002] direction are displayed in **a** and in **b**–**c**, respectively. Note that all potentials were converted to the RHE scale and 100% iR-compensated.

the particle's shape, the 3D Bragg peak intensity is analyzed and complex variations in the electron density are required to be precisely recovered. The displacement field can be derived from the phase of the complex electronic density map of an individual NP. Computational efforts, e.g., iterative phase retrieval algorithms[27,28], are utilized to reconstruct the particle shape as well as 3D strain maps[29,30].

Regarding the acquired potential-dependent diffraction patterns (Fig. 4), slight changes can be distinguished for the scan at −1.10 V$_{RHE}$ as well as the subsequent scan at OCP. Even though variations in the averaged lattice constants are below the resolution limit (<10$^{-3}$%), we discover slight variations in the structure of the NP for negative potentials of ~−1.10 V$_{RHE}$. Importantly, this effect seems to be irreversible, as the fringe pattern at the subsequent OCP step remains distorted.

Note that profound investigations of multiple NPs on the electrodes revealed the presence of different crystal orientations, with the majority of NPs exhibiting (111) and (002) orientations. Figure 5a illustrates the potential-dependent local deformation field inside an individual NP. The displacement field projection (Fig. 5a) and the local strain maps (Fig. 5b) along the [002] direction highlight the presence of tensile strain at the top and the bottom (i.e., at the interface to the GC substrate) of the crystal. At the center, contrarily, pronounced compressive strain is found. Between −0.85 and −1.00 V$_{RHE}$, the overall shape of the crystal remains essentially unaltered. Yet, compared to the initial reconstructed crystal, an increase in compressive and tensile

strain is found at the center and the top, respectively (Fig. 5b). Between −1.05 and −1.10 V$_{RHE}$, a contraction in particle height, i.e., perpendicular to the substrate, can be observed (Fig. 5a, b, top/bottom). Beyond that, notch formation featuring reduced tensile strain is evident at the interface to the substrate at −1.05 V$_{RHE}$ (Fig. 5c, top/bottom). Concomitantly, a reduced tensile strain is found at the top of the crystal (Fig. 5b, top/bottom). At −1.10 V$_{RHE}$, tensile strain increases at the top and the overall shape appears stretched along the x-axis. Tensile strain is predominant in the vicinity of the potential-induced notches at both the top and the bottom of the crystal, where the formation of local stress maxima is evident (see Fig. 5b, middle/bottom). Both the diffraction pattern (Fig. 4) and the strain map (Fig. 5b, c) derived from the last scan at OCP differ slightly from that of the initial state. Most notably, a diagonal strain propagation towards the surface of the crystal can be observed. Beyond that, the crystal irreversibly changed in overall shape and height.

## Discussion

Ligand-free, bimetallic Cu-Ag NPs serve as robust model eCO$_2$RR catalysts for the assessment of product selectivity and active site(s) identification. As a poor HER catalyst, the presence of low amounts of Ag atoms is found to effectively reduce competing H$_2$ formation, which can be attributed to its larger free energy of adsorption relative to that of Cu[18]. Using hard X-ray-based nanoprobe techniques imposes strict requirements on the electrochemical setup and catalysts/

electrodes (see Supplementary Figs. 8 and 14 for further information). The combination of nano-XRF and nano-XAS represents an emerging combination of advanced methodologies to probe local differences in the average oxidation state of an electrocatalyst and distinguish between a specific vs. an ensemble response. We note that the obtained information mainly originates from the bulk of the nanometer-scale particles. Clark and co-workers[4] claimed the formation of Cu-Ag surface alloys with low Ag content and an enhanced selectivity toward oxygenates. Compressively-strained, less oxophilic Cu surface species were hypothesized. Rejecting the CO spillover concept, the authors conjectured surface strain as the main parameter to tune selectivity and, more generally, to explain the synergistic effects between Cu and Ag. The present contribution now provides direct evidence for three-dimensional atomic displacement/strain effects in bimetallic model catalysts with distinct Cu and Ag domains.

Nano-BCDI offers valuable information about the intraparticle strain distribution[31]. Currently, there are only a few reports about in situ BCDI studies. Recent advances improved the resolution as well as the coherent photon flux, which favorably reduced the acquisition time[30]. Taking advantage of the fourth generation Extremely Brilliant Source of the ESRF and a nanoprobe end-station, detailed insights into the dynamics of strain evolution of nanomaterials can be obtained[32]. Recently, a detailed study[33] proposed a methodology for a reliable, quantitative analysis of displacement field and strain in nanocrystals. As illustrated in Fig. 5c, the maximum intraparticle strain along the [002] direction stayed within ±0.1% over the entire investigated potential range. Ex situ EDX elemental (Fig. 2g, h and Supplementary Figs. 2 and 3) and XRD analyses (Fig. 2i and Supplementary Fig. 5) evidence a segregated structure of Cu and Ag domains. Note that a polycrystalline character is predominant, as indicated by TEM/SAED analysis (Supplementary Fig. 3). Contrary to the evaluation of lattice strain by XRD analysis (ensemble response), nano-BCDI provides spatially resolved information (specific response) with a significantly improved accuracy, particularly for NP-based electrodes.

A few durable, Ag-rich NPs are present in both $Cu_{0.88}Ag_{0.12}$ and $Cu_{0.95}Ag_{0.05}$ samples (see Supplementary Fig. 6), which can be rationalized based on the Cu-Ag phase diagram[34]. These Ag-rich NPs, however, could not be properly analyzed via nano-XAS (see Supplementary Fig. 9 for further information). An averaged lattice constant of 4.074 Å was derived from the nano-BCDI data, which, according to Vegard's law, indicates an Ag content of ~97 at-% for these few Ag-rich NPs with (002) orientation. Overall, sample stability remains the main challenge for a reliable in situ nano-BCDI evaluation under $eCO_2RR$ conditions. In this context, $Cu_{0.88}Ag_{0.12}$-based electrodes feature improved particle stability under X-ray irradiation, enabling full reconstruction of an individual crystal (Fig. 5).

In brief, for $Cu_{0.88}Ag_{0.12}$-based electrodes, most NPs reveal a robust, Janus-type structure with distinct metal domains and fully reduced (i.e., metallic) chemical states (cf. Supplementary Fig. 13), leading not only to an improved product selectivity but also to an enhanced durability. At about −0.95 $V_{RHE}$, an increased CO Faradaic efficiency (FE) is found, which can either be attributed to the presence of the Ag domain or to the intraparticle Cu-Ag interface (see Supplementary Fig. 7)[5,35]. The $C_1$ product selectivity is significantly higher for bimetallic catalysts compared to monometallic Cu over the entire investigated potential range. The inherent local atomic displacement and lattice strain inside an individual NP appear to be key in affecting the $eCO_2RR$ selectivity. At moderately negative potentials below −1.00 $V_{RHE}$, the overall shape of a bimetallic NP undergoes subtle changes compared to the initial reconstructed crystal (Fig. 5). Most notably, notch formation is observed at the interface to the substrate at more negative potentials of −1.05 $V_{RHE}$ (Fig. 5c, bottom) which results in an uneven stress distribution accompanied by a rise in $CH_4$ FE (see Supplementary Fig. 7). Simultaneously, an ongoing, irreversible contraction in particle height is

found. Even at very negative potentials, the $C_2H_4$ FE remained fairly low, which can be addressed to the relatively broad spatial Cu/Ag transition region (Fig. 2b–d and Supplementary Figs. 2, 3), inhibiting an effective transfer of CO species from Ag to Cu as well as concomitant CO dimerization on rough, oxide-derived (OD) Cu domains[36]. Indeed, NPs featuring sharp Cu/Ag interfaces were previously designed to promote interfacial charge transfer and increase $C_2H_4$ selectivity[37–39]. Herein, the presence of the Ag domains effectively impedes the evolution of rough, OD-Cu domains that were found to boost CO dimerization. For the $Cu_{0.88}Ag_{0.12}$ model catalyst, a similar trend in product selectivity as recently reported by Choi et al. [39] for nanowires with an almost identical composition is found. At −1.10 $V_{RHE}$, a change in diffraction pattern (Fig. 4) occurs, which provides evidence for a potential-induced restructuring of the crystal, accompanied by a further enhanced $CH_4$ selectivity. Negative (cathodic) potentials thus foster the evolution of local strain maxima −both tensile and compressive. An irreversible, diagonal strain/stress propagation toward the particle's surface is directly visualized (Fig. 5b, c, bottom).

According to the findings from the reconstruction (Fig. 5a–c), reaction-induced eigenstress[7] seems less relevant for potentials more positive (anodic) than −1.05 $V_{RHE}$. Considering the segregated domain structure, the lattice mismatch between macroscopic Cu and Ag domains is proposed to induce interfacial tensile strain on the Cu lattice. Albeit no direct evidence for the exact location of the Ag domain can be given, likely, the region with the highest tensile strain (i.e. the top of the crystal) represents the Cu-Ag interface. Additionally, we hypothesize preferential adsorption at the top of the bimetallic NP. For strongly negative (cathodic) potentials, the crystal's contraction along the z-axis most likely results from dynamic restructuring processes, as indicated by slight variations in the potential-dependent fringe patterns (Fig. 4, red arrows). Despite their segregated structure, the presence of the Ag domain and the intraparticle Cu-Ag interface can significantly reduce undesired dissolution-redeposition processes under $eCO_2RR$ conditions, as previously described in one of our studies[5]. Cu-based NPs, containing only minute amounts of Ag, reveal delicate properties thwarting a full reconstruction of individual crystals (see Supplementary Fig. 15 for further details).

We have introduced, validated, and utilized a powerful methodology to directly visualize the presence and the dynamics of heterogeneous (both compressive and tensile) lattice strain in individual, catalytically operating, bimetallic Cu-Ag NP catalysts. Due to the immiscibility of Cu and Ag, segregated Janus-type NPs with distinct Cu and Ag domains were obtained after dewetting at high temperatures in a reductive atmosphere. In situ, nano-XRF/-XAS experiments provided evidence for a fully reduced average oxidation state (0) for both metal species in the bulk under $eCO_2RR$ operating conditions in $CO_2$-saturated $KHCO_3$ electrolyte. In situ nano-BCDI experiments shed light on the strain distribution in individual NPs, revealing a potential-dependent restructuring. Tensile and compressive strain in the range of ±0.1% are present in the initial state, undergoing subtle changes for more negative (cathodic) potentials than −1.05 $V_{RHE}$. Irreversible contraction of the crystal as well as notch formation were observed, which correlate with an enhanced $CH_4$ selectivity, while $C_2H_4$ formation is suppressed by the segregated structure without a sharp Cu/Ag interface enabling fast CO spillover and C–C coupling. This study highlights the immensely complex lattice dynamics of individual, bimetallic, nanometer-sized catalyst particles. It provides direct evidence for synergistic effects between Cu and Ag species significantly promoting $C_1$ product selectivity and catalyst durability. We expect the present combination of in situ X-ray nanoprobe techniques to be utilized more broadly in the NP catalysis community to unravel the roles of lattice strain and redox states in electrocatalysis.

## Methods

### Sample preparation and characterization

Monometallic Cu and bimetallic Cu-Ag NPs were synthesized via sputter deposition of thin films onto polished GC rods. To control the composition of each catalyst material, the deposition time of Ag was varied (0 s, 5 s, 10 s, 30 s, and 60 s). Before the deposition of Ag, 60 s of Cu sputter-coating was conducted. The obtained thin films were subsequently dewetted by thermal treatment for 6 h at 750 °C in a mixture of $H_2$ in Ar (4 vol-% $H_2$). SEM and SEM-EDX analyses were carried out on a JEOL JSM-7401F operated at 10 kV. GI-XRD analysis was conducted on a Bruker D8 Advance instrument with an angle of 0.3° for the incident beam and Cu-$K_\alpha$ radiation. The preparation of a lamella for TEM analysis was carried out at a FEI Helios NanoLab 600 DualBeam SEM/FIB. After lift-out, Ga ion beams of ~2.80, 0.28, and 0.09 nA (30 kV) were used to polish the lamella to electron transparency at 3 keV. Finally, the lamella was showered with 2 keV ions at ~28 pA and ±5° incidence angle to reduce the surface amorphization. The total dose applied was 268 pC $\mu m^{-2}$ on each side. SAED analysis was conducted on a Tecnai G2 20 S−TWIN operated at 200 kV. Using an aperture, the investigated sample area was restricted to a spherical area with a diameter of ~130 nm.

### Electrochemical characterization

For an initial assessment of the $eCO_2RR$ durability of the mono and bimetallic NP based electrodes, consecutive chronoamperometry measurements were conducted at (uncompensated) potentials of -0.90, -1.00, -1.05, -1.10, -1.15 and -1.20 $V_{RHE}$. The duration of each potential step was at least 60 min. Measurements were conducted in aqueous 0.1 M $KHCO_3$ solution as a supporting electrolyte (pH 6.8) and at room temperature. For a determination of the potential-dependent selectivity of the different catalysts with varying compositions, gaseous and liquid $eCO_2RR$ products were analyzed via GC and HPLC, respectively. More details about the electrochemical testing can be found elsewhere[40].

### In situ nanofocused spectroscopy

Nanofocused in situ XAS experiments were performed at the ID16B beamline at ESRF, France, using hard X-rays (over 5–10 keV) focused to a spot size of ~80 × 100 nm² (H × V FWHM). XANES data were first normalized and then analyzed using a customized Python script and the Athena software, respectively, as described elsewhere[41]. Scans were acquired from ~8.93 to 9.13 keV. For the XRF maps, raster scans of ~3 × 3 µm² step sizes with a resolution of 0.1 × 0.1 µm² were carried out using an exposure time of 100 ms. Nanofocused in situ BCDI experiments were performed at the P10 coherence applications beamline at DESY, Germany, using an X-ray energy of 13.095 keV with an energy resolution of 1.12 eV and a bandwidth of $\delta\lambda/\lambda \approx 1.4 \times 10^{-4}$. The X-ray beam was focused to a size of ~1.2 µm × 0.85 µm (H × V FWHM). BCDI diffraction data were processed using two Python packages: PyNX[27] for phase retrieval and cdiutils[42] for pre-and post-processing. The phase retrieval involved several steps, including a series of 1000 Relaxed Averaged Alternating Reflections (RAAR[43]), 200 Hybrid Input/Output steps (HIO[28]) and 150 Error-Reduction (ER[44]) steps. The phasing process included a partial coherence algorithm to account for the partially incoherent incoming wave front[45]. To ensure the best reconstruction possible, we first selected the ten best reconstructions (with the lowest free log-likelihood[46]) from 30 with random phase starts. Then, five reconstructions were selected from these ten by selecting the reconstructions with the lowest value of the absolute difference between the maximum and averaged electron density (mean-to-max criterion in the cdiutils package). Finally, we performed the decomposition into modes from the last five reconstructions[46]. In the post-processing stage, the reconstructed object was interpolated onto an orthogonal grid (laboratory frame) for ease of visualization. Isosurface determination and phase unwrapping were then performed. Phase shift

adjustments were made to ensure the averaged phase over the isosurface-bounded reconstruction was zero. The code and methods used in this process are available elsewhere[42].

For both synchrotron techniques, a custom-made in situ cell was used. Freshly prepared $CO_2$-saturated 0.1 M $KHCO_3$ electrolyte solution was continuously pumped through the setup (1 mL $min^{-1}$). The working electrode (WE) was located at the center of the cell, which was sealed using a thin (~6 µm) Mylar film. For an optimized cell pressure, the height of the electrolyte reservoir relative to the mounted cell was adjusted. During the in situ experiments under negative potentials, the formation of large gas bubbles was observed at the electrode's surface. To mitigate extensive bubble formation, OCP periods in the range of several minutes were inserted after each potential step. Note that both the formation and the removal of large gas bubbles were found to interfere with the acquisition of the XAS data.

## Data availability

All data supporting the results of this study are included in the published article or the associated Supplementary Information.

## Code availability

The code used can be found in refs. 27,42.

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

## Acknowledgements

The authors thankfully acknowledge financial support from the European Union's Horizon 2020 research and innovation funding program under grant agreements no. 851441 (SELECTCO2) and no. 101006701 (EcoFuel), as well as from the Deutsche Forschungsgemeinschaft (German Research Foundation, DFG) under grant no. STR 596/18–1. M.F. and A.A. thankfully acknowledge financial support by the Federal Ministry of Education and Research (Bundesministerium fuer Bildung und Forschung, BMBF) under grant no. 03SF0611A (H2Meer). L.W. gratefully acknowledges financial support from the Alexander von Humboldt Foundation. C.A. and M.-I.R. thankfully acknowledge financial support from the European Research Council (ERC) under the European Union's Horizon 2020 research and innovation program (grant agreement no. 818823, CARINE). H.N.N. gratefully acknowledges funding from the DFG under grant no. STR 596/21-1 (DaCapo). We acknowledge the European Synchrotron Radiation Facility (ESRF) for provision of synchrotron radiation facilities using beamlines ID01 and ID16B. We also acknowledge DESY (Hamburg, Germany), a member of the Helmholtz Association HGF, for the provision of experimental facilities. Parts of this research were carried out at PETRA III using the P10 Coherence Applications Beamline. Beamtime was allocated for proposal I-20220139. We also thank Dipl.-Ing. Sören Selve and Dr. Christian Günther from the Zentraleinrichtung für Elektronenmikroskopie (ZELMI) at Technische Universitaet Berlin (TUB) for their support with FIB/SEM lamella preparation as well as STEM-EDX mappings. In addition, the authors thank Dr. Malte Klingenhof and Paul W. Buchheister (TUB) for their support with additional BCDI experiments carried out at the ID01 beamline at ESRF.

## Author contributions

M.F., L.W., C.A., M.-I.R., and P.S. designed the experiments and developed the concept of the study. M.F., L.W., M.H., R.T., L.L., J.L., and J.V. carried out in situ nano-XAS experiments, evaluated and discussed the data. M.F.,

L.W., C.A., Z.R., A.G., H.N.N., M.S., and M.-I.R conducted in situ nano-BCDI experiments, evaluated and discussed the data. M.F. and L.W. carried out SEM and SEM/EDX analyses; M.F. and A.A. carried out TEM/SAED analysis after FIB/SEM lamella preparation and evaluated the data. L.W. performed the electrochemical tests of the eCO2RR catalysts. All authors contributed to scientific discussions and jointly revised the manuscript.

## Funding

## Competing interests
The authors declare no competing interests.
