## [Peer Review File · Nature Communications]

REVIEWER COMMENTS

Reviewer #1 (Remarks to the Author):

In this manuscript, the authors reported the in-situ nanoprobe techniques for the investigation of reaction mechanisms for Cu-Ag nanoparticles towards CO₂RR. This work is of significance for the developing of in-situ characterizations for reaction mechanisms and could be recommended for publication before the following issues be addressed.

- (1) The XANES spectra of Cu displayed comprehensive evidence of the structural evolution of Cu in CuAg nanoparticles. How about the structural evolution of Ag? Why the authors didn't collect the XANES spectra of Ag?
- (2) The structure and strain information were discussed with the application of in situ characterizations. The authors should offer more discussion about the structure evolution and strain under the same potential, such as -1.05 and -1.1 V.
- (3) The significant change in XANES spectrum could be observed in Figure S11, which may not sincerely due to the change of chemical states.
- (4) The coordination environment of Cu is suggested to be analyzed if possible.

Reviewer #2 (Remarks to the Author):

The authors describe the synthesis of Cu-Ag tandem nanoparticle catalysts for electrochemical CO₂ reduction, and their characterization with in-situ nanofocused X-ray absorption spectroscopy (nano-XAS) and nanofocused Bragg coherent diffraction imaging (nano-BCDI). These methods allow for spectroscopic and structural analysis of individual nanoparticles, with a focus on evolving strain within the particles as the reaction proceeds. This is motivated by literature studies suggesting that strain engineering may provide an effective means of catalyst design, but strain effects are difficult to directly observe.

The Cu-Ag nanoparticles are shown to be phase segregated rather than alloyed (i.e. Janus particles), with small areas of Ag attached to primarily Cu particles. These show impressive stability, as demonstrated with a direct comparison to pure Cu catalysts, even though Ag content is small (12% for the best performing catalyst). This stability allows the authors to interrupt catalysis with periods of open circuit potential (OCP) to align measurements and identify individual nanoparticles to study. The results are impressive, particularly for the nano-BCDI measurements, which yield 2D and 3D reconstructed strain maps of individual particles operating at different reductive potentials, before a final return to

OCP. These reveal regions of tensile and compressive strain in the particles, aligned along the “vertical” axis (I assume this means normal to the substrate? But it should be clarified), and the strain evolves across the different potentials studied. Specifically, the particles become more compressed in the vertical axis, with more tensile strain at the top and compressive strain in the middle, at more reductive potentials. Much of this restructuring is irreversible. The authors hypothesize that the Cu-Ag interface induces tensile strain on the Cu lattice, with preferential adsorption of reactive intermediates on the top, and a dynamic restructuring process at the more reductive potentials.

Overall the study is impressive and worthy of publication, but I do have a few comments:

1) The authors are unable to track the Ag domains specifically in the in situ characterizations. They cite interference from the Ar (in air) and K (in the electrolyte) K-edge absorption lines as the reason that Ag L-edge XAS could not be conducted. However, these interferences do not exist for the Ag K-edge (at 25.5 keV). Were the beamlines used for the study unable to reach this edge? The authors admit that Ag K-edge studies would be informative, but I assume infeasible for inclusion in this study? Are the authors planning follow-up measurements using Ag K-edge XAS? This is relevant for my second comment as well...

2) The authors offer little explanation for the changes observed in the nanoparticles during reaction. Without the Ag K-edge XAS it is difficult to make concrete conclusions, but I would have appreciated some possible mechanisms for the observed changes, and particularly how the Ag incorporation achieves such high stability compared to the pure Cu catalysts.

Some other minor comments also:

- Does Figure 3c,d show XANES from a single nanoparticle, or some sort of average? Supp. Fig. 10 specifically states it is tracking individual particles, but Fig. 3 does not. Additionally, can the authors comment on small differences in spectral shape across the different potentials shown (which also appear in Supp. Fig. 10)? I agree that the spectra appear to be metallic Cu in all cases, but some have better resolved features than others, almost as if there is a varying degree of overabsorption occurring (although that doesn't make sense given the nanoscale catalysts studied here). Why this variance in spectra?

- Pure Cu catalysts have been shown to rapidly oxidize when returning to OCP (for example, <https://doi.org/10.1021/jacs.0c10017>). The authors return to OCP often in between their measurements, do they see any evidence of Cu oxidation during these periods? This should be clear in the Cu K-edge XAS if it occurs. It may affect particle structures between applied potential runs. If it does not occur, can the authors hypothesize why the incorporation of Ag would prevent Cu reoxidation?

- Fig. 5, "Z is along the particle's height", does this mean Z is normal to the glassy carbon substrate? This should be clarified throughout the text so the explanations about the "top" of the particles makes sense to readers.

Response to reviewers' comments

Reviewer comments:

Reviewer # 1

In this manuscript, the authors reported the in-situ nanoprobe techniques for the investigation of reaction mechanisms for Cu-Ag nanoparticles towards CO₂RR. This work is of significance for the developing of in-situ characterizations for reaction mechanisms and could be recommended for publication before the following issues be addressed.

Our answer:

We thank the referee for the overall positive assessment of our work.

(1) The XANES spectra of Cu displayed comprehensive evidence of the structural evolution of Cu in CuAg nanoparticles. How about the structural evolution of Ag? Why the authors didn't collect the XANES spectra of Ag?

Our answer:

First of all, we appreciate the referee's remark. This aspect is also mentioned by the second referee. Although Ag K-edge could be reached at ID16B beamline at the ESRF, in principle, the relatively low Ag concentration makes it very challenging to acquire nanofocused Ag XANES with a sufficient signal-to-noise ratio. An increased energy of the incoming X-ray beam for Ag K-edge spectroscopy (> 25 keV) can be expected to foster beam damage. Considering the nanofocused geometry, extensive local heating issues would impede reliable *in situ* measurements, amongst others. During our experiments at the beamline, the acquisition of reproducible Cu K-edge XANES under *in situ* conditions has already been quite challenging. Note that there are currently only a very few reports about successful nanofocused XAS measurements under operating conditions. Yet, we are convinced that future improvements in nanoprobe techniques will enable meaningful characterizations of more complex nanoscale (electro-)catalysts.

(2) The structure and strain information were discussed with the application of in situ characterizations. The authors should offer more discussion about the structure evolution and strain under the same potential, such as -1.05 and -1.1 V.

Our answer and actions:

We thank the referee for his remark and do now provide additional data from our *in situ* experiments in the SI (page 14, Supplementary Fig. 13). A reference has been added in the main part (page 9, top). In brief, only small variations in the electronic structure were found according to the evaluation of the acquired nano-XAS data during chronoamperometric measurements at negative potentials. As a representative example, we added further spectra revealing that metallic Cu⁽⁰⁾ remain the predominant species for consecutive measurements at -1.15 and -1.25 V_{RHE} in the range of several minutes. Regarding the dynamic evolution of strain in individual crystals, we could not find any indications for significant variations in the

diffraction fringes at a specific potential. A reliable analysis of the dynamic evolution of strain inside individual crystals on larger time scales is highly challenging and lies beyond the scope of our current work. We admit that we faced several particle (in)stability issues during our *in situ* BCDI experiments and further improvements in electrode design would be necessary in order to run extended measurement durations.

(3) The significant change in XANES spectrum could be observed in Figure S11, which may not necessarily be due to the change of chemical states.

Our answer and actions:

We thank the referee for his remark. In Supplementary Fig. 11, nano-XAS evaluations for Cu species close to the particle rim (i.e. close to the electrolyte interface) are given and discussed. Overall, the signal-to-noise ratio is lower compared to the spectra obtained from the center of individual NPs (e.g. Supplementary Fig. 10) due to a lower number of atoms being analyzed. This may indeed lead to more pronounced variations in the shape of the spectra. We added further hypotheses to the SI (page 12). Furthermore, oxidized Cu^{II} species at the particle's surface certainly show a different structure and coordination compared to metallic Cu⁽⁰⁾ in the bulk. Unfortunately, reliable EXAFS analysis using the nanoprobe technique has not (yet) been feasible (*vide infra* for further explanations).

(4) The coordination environment of Cu is suggested to be analyzed if possible.

Our answer:

We appreciate the referee's suggestion to add EXAFS data to our study. However, due to the nanofocused technique applied herein, a reliable fitting in the EXAFS range was not possible. The nanoprobe approach certainly leads to a tradeoff in XAS signal quality. Additionally, the formation of gas bubbles at the working electrode affected the post-edge region. In the present work, the main focus lies on the evaluation of oxidation states of Cu species. In this context, the nanofocused geometry was successfully employed in order to gain distinct insights into the electronic properties of bulk and - at least to some extent - near-surface Cu species in individual, bimetallic NPs.

Reviewer #2

The authors describe the synthesis of Cu-Ag tandem nanoparticle catalysts for electrochemical CO₂ reduction, and their characterization with in-situ nanofocused X-ray absorption spectroscopy (nano-XAS) and nanofocused Bragg coherent diffraction imaging (nano-BCDI). These methods allow for spectroscopic and structural analysis of individual nanoparticles, with a focus on evolving strain within the particles as the reaction proceeds. This is motivated by literature studies suggesting that strain engineering may provide an effective means of catalyst design, but strain effects are difficult to directly observe.

The Cu-Ag nanoparticles are shown to be phase segregated rather than alloyed (i.e. Janus particles), with small areas of Ag attached to primarily Cu particles. These show impressive stability, as demonstrated with a direct comparison to pure Cu catalysts, even though Ag content is small (12% for the best performing catalyst). This stability allows the authors to interrupt catalysis with periods of open circuit potential (OCP) to align measurements and identify individual nanoparticles to study. The results are impressive, particularly for the nano-BCDI measurements, which yield 2D and 3D reconstructed strain maps of individual particles operating at different reductive potentials, before a final return to OCP. These reveal regions of tensile and compressive strain in the particles, aligned along the “vertical” axis (I assume this means normal to the substrate? But it should be clarified), and the strain evolves across the different potentials studied. Specifically, the particles become more compressed in the vertical axis, with more tensile strain at the top and compressive strain in the middle, at more reductive potentials. Much of this restructuring is irreversible. The authors hypothesize that the Cu-Ag interface induces tensile strain on the Cu lattice, with preferential adsorption of reactive intermediates on the top, and a dynamic restructuring process at the more reductive potentials.

Overall the study is impressive and worthy of publication, but I do have a few comments:

Our answer:

We thank the referee for the overall very positive assessment of our work.

1) The authors are unable to track the Ag domains specifically in the in situ characterizations. They cite interference from the Ar (in air) and K (in the electrolyte) K-edge absorption lines as the reason that Ag L-edge XAS could not be conducted. However, these interferences do not exist for the Ag K-edge (at 25.5 keV). Were the beamlines used for the study unable to reach this edge? The authors admit that Ag K-edge studies would be informative, but I assume infeasible for inclusion in this study? Are the authors planning follow-up measurements using Ag K-edge XAS? This is relevant for my second comment as well...

Our answer:

We thank the referee for his remark. This aspect was also mentioned by the first referee. Although Ag K-edge could be reached at ID16B beamline at the ESRF, in principle, the relatively low Ag concentration makes it very challenging to acquire nanofocused Ag XANES with a sufficient signal-to-noise ratio. An increased energy of the incoming X-ray beam for Ag K-edge spectroscopy (> 25 keV) can be expected to foster beam damage. Considering the nanofocused geometry, extensive local heating issues would impede reliable *in situ* measurements, amongst others. During our experiments, the acquisition of reproducible Cu K-edge XANES under *in situ* conditions has already been quite challenging. Note that there

are currently only a very few reports about successful nanofocused XAS measurements under operating conditions. Yet, we are convinced that future improvements in nanoprobe techniques will enable meaningful characterizations of more complex nanoscale (electro)catalysts.

2) The authors offer little explanation for the changes observed in the nanoparticles during reaction. Without the Ag K-edge XAS it is difficult to make concrete conclusions, but I would have appreciated some possible mechanisms for the observed changes, and particularly how the Ag incorporation achieves such high stability compared to the pure Cu catalysts.

Our answer and actions:

We thank the referee for his comment. In our previous study on Cu-Ag tandem catalysts, we have already found evidence for a synergistic effect between adjacent Cu and Ag species (DOI: 10.1039/D0NR09040A). Although synthesized through different approaches, segregated domains of Cu and Ag were reproduced in the present study. As such, the immiscibility of Cu and Ag species can be confirmed to play a major role in inhibiting unfavorable dissolution-redeposition process of nanoscale eCO₂RR catalyst materials, as outlined before. We rephrased the corresponding section on **page 9 (bottom) in the main part** of our work.

Some other minor comments also:

- Does Figure 3c,d show XANES from a single nanoparticle, or some sort of average? Supp. Fig. 10 specifically states it is tracking individual particles, but Fig. 3 does not. Additionally, can the authors comment on small differences in spectral shape across the different potentials shown (which also appear in Supp. Fig. 10)? I agree that the spectra appear to be metallic Cu in all cases, but some have better resolved features than others, almost as if there is a varying degree of overabsorption occurring (although that doesn't make sense given the nanoscale catalysts studied here). Why this variance in spectra?

Our answer and actions:

We thank the referee for his comment. Indeed, all shown XANES spectra were acquired on individual NPs. We added further details to the **caption of Fig. 3 on page 5 in the main part**. The variance in the nano-XAS spectra has also been mentioned by the first reviewer. Briefly, we are convinced that the subtle variations in spectral shape can be attributed to the nanoprobe technique applied in our study. Both particle and beam movements during the measurement may affect the spectral shape/features. Beyond that, gas bubble formation and local heating at the working electrode could play a role. Nevertheless, we are convinced that the observed small spectral variations do not significantly alter our main findings or conclusions.

- Pure Cu catalysts have been shown to rapidly oxidize when returning to OCP (for example, <https://doi.org/10.1021/jacs.0c10017>). The authors return to OCP often in between their measurements, do they see any evidence of Cu oxidation during these periods? This should be clear in the Cu K-edge XAS if it occurs. It may affect particle structures between applied potential runs. If it does not occur, can the authors hypothesize why the incorporation of Ag would prevent Cu reoxidation?

Our answer and actions:

We appreciate the referee's remark. We added additional nano-XAS data to the **SI of our work on page 14**, in which an individual NP had been investigated more thoroughly. Several spectra were recorded at OCP, i.e. prior to the step at $-1.15 V_{RHE}$ as well as after the step at $-1.25 V_{RHE}$ and in between, all revealing similar features and shape. Accordingly, at the time scale of our measurements, no significant indications for any pronounced Cu (re-)oxidation were found. This, in turn, is claimed to improve the catalytic stability of the bimetallic NPs.

- Fig. 5, "Z is along the particle's height", does this mean Z is normal to the glassy carbon substrate? This should be clarified throughout the text so the explanations about the "top" of the particles makes sense to readers.

Our answer and actions:

We thank the referee for his comment and rephrased the corresponding section in the main part of our work in order to improve clarity (**page 7, caption of Fig. 5**).

REVIEWERS' COMMENTS

Reviewer #1 (Remarks to the Author):

The revision is improved and suitable for publishing, no more comments from my side.

Reviewer #2 (Remarks to the Author):

The authors have addressed my concerns, and I am happy to recommend publication. The only edit I would suggest is in the new Supplementary Fig. 13... in the authors' reply to reviewer comments, they mention that several XANES spectra were collected at OCP in between the applied potentials, but only one OCP spectrum is shown. Adding more to the figure would make it clear to readers that no re-oxidation occurs when returning to OCP during the measurement.

I'd also like to point out one error in the authors' explanation for not conducting Ag K-edge measurements: the authors argue that at the higher X-ray energy of 25.5 keV, there will be more beam damage on the sample. I would expect the opposite. At the higher X-ray energy the sample will be generally less absorbing and therefore less susceptible to beam damage, unless the authors anticipate damage to the Ag domains specifically due to resonant absorption. The other factors mentioned by the authors, such as low Ag concentrations and general challenges related to the nanofocused experiment, are more convincing, and I concede that Ag K-edg measurements would be infeasible to include with this publication.

Response to reviewers' comments (2)

Unraveling the Synergistic Effects of Cu-Ag Tandem Catalysts during Electrochemical CO₂ Reduction using Nanofocused X-ray Probes

We highly appreciate the reviewers' comments on our contribution. In the following, we provide a detailed point-by-point response to their remarks. All text changes in the main part as well as in the SI were highlighted in yellow.

Reviewer comments:

Reviewer #1

The revision is improved and suitable for publishing, no more comments from my side.

Our answer:

We thank the referee for the positive assessment of our previous manuscript version.

Reviewer #2

The authors have addressed my concerns, and I am happy to recommend publication. The only edit I would suggest is in the new Supplementary Fig. 13... in the authors' reply to reviewer comments, they mention that several XANES spectra were collected at OCP in between the applied potentials, but only one OCP spectrum is shown. Adding more to the figure would make it clear to readers that no re-oxidation occurs when returning to OCP during the measurement.

Our answer & actions:

We thank the referee for the positive assessment of our revised manuscript and agree with his remark. Accordingly, we added an additional spectrum collected at OCP, as well as further explanations, to Supplementary Figure 13 (a, b) in order to emphasize that no re-oxidation of bulk Cu species was observed. Note that we are aware that surface re-oxidation processes are likely to occur when switching from negative potentials to OCP. However, more surface-sensitive characterization techniques would be required to investigate the redox chemistry of surface species.

I'd also like to point out one error in the authors' explanation for not conducting Ag K-edge measurements: the authors argue that at the higher X-ray energy of 25.5 keV, there will be more beam damage on the sample. I would expect the opposite. At the higher X-ray energy the sample will be generally less absorbing and therefore less susceptible to beam damage, unless the authors anticipate damage to the Ag domains specifically due to resonant absorption. The other factors mentioned by the authors, such as low Ag concentrations and general challenges related to the nanofocused experiment, are more convincing, and I concede that Ag K-edge measurements would be infeasible to include with this publication.

Our answer and actions:

We acknowledge the referee's comment and agree. We had placed focus on other factors, such as low Ag concentrations and general challenges related to the nanofocused experiment, rather than beam damage. We have carefully removed any erroneous assumptions from our work regarding the adverse effect of an increased beam energy on particle stability.

We would like to add that, in general, varying the X-ray energy requires careful optimization of several parameters, such as dose rate, profile of the beam and sensitivity of the detector. In the present study, these optimizations were not feasible due to time constraints at the beamline. We are convinced that further investigations regarding the environment of the Ag species are essential to fully understand the proposed synergistic effects of the tandem catalyst particles.